# A workflow to visualize vertebrate eyes in 3D

**Jiayun Wang**, **Sabine Baumgarten, Frederic Balcewicz, Sandra Johnen, Peter Walter***,
**Tibor Lohmann**

Department of Ophthalmology, RWTH Aachen University, Aachen, Germany

* pwalter@ukaachen.de

## Abstract

### Purpose

To establish a workflow to visualize the surgical anatomy in 3D based on histological data of eyes of experimental animals for improving the planning of complex surgical procedures.

### Methods

Four C57BL/6J wild-type(wt) mouse eyes, three Brown Norway rat eyes and four Chinchilla Bastard rabbit eyes were enucleated and processed for standard histology with serial sections and hematoxylin and eosin staining. Image stacks were processed to obtain a representation of the eye anatomy in 3D. In addition, virtual image stacks and 3D point clouds were generated by processing sagittal sections of eyes with stepwise 180˚ rotation and projection around the eye axis to construct a rotationally symmetric 3D model from one single sagittal section.

### Results

Serial sections of whole eyes of mice, rats and rabbits showed significant artifacts interfering with a practical image stack generation and straightforward 3D reconstruction despite the application of image registration techniques. A workflow was established to obtain a 3D image of the eye based on virtual image stacks and point cloud generation by rotation of a single sagittal section of the eye around the symmetry axis. By analyzing the tissue shrinkage during histological processing true biometric reconstructions of the eyes were feasible making the resulting model usable for 3D modeling and simulation, e.g. for planning of complex surgical procedures in different species.

### Conclusion

Because serial sections of the eye with standard histological protocols yielded too many artifacts for a straightforward 3D visualization we reconstructed a pseudorealistic 3D model based on virtual image stacks and point cloud generation calculated from a single sagittal section of the eye. Such a model detailing microscopic structures of the whole eye will allow for a specific planning of surgical procedures in small animal eyes in order to prevent surgical complications in a very early stage of an experiment and it will support the design and development of complex intraocular implants. It will therefore be helpful in surgical teaching

**Data Availability Statement:** All relevant data are within the manuscript.

**Funding:** The project is part of a graduate school funded by the Deutsche Forschungsgemeinschaft DFG under GRK 2610/1. All authors are supported

by this grant. The funders had no role in study design, data collection and analysis, decision to publish, or preparation of the manuscript.

and improve laboratory animal welfare by an expected reduction of experimental animal numbers. Further processing including integration of mechanical tissue properties is needed to convert these 3D models into a practical virtual reality teaching and simulation platform for eyes of several species.

## Introduction

For planning and performing of complex surgical procedures in eyes of laboratory animals, a deep understanding of the anatomy of these eyes is mandatory. Although the eye anatomy in species such as mice, rats, or rabbits is well described, modern visualization tools may support surgeons, researchers, and instrument designers to improve their performance. In situations where complex and large devices consisting of different materials and substructures or modules must be implanted and fixated in the eye, the number and severity of complications may be significant. One field of application is the development and implantation of devices used for electrical stimulation of the retina (retinal implants). In studies with new experimental devices implanted in rabbit eyes we realized that retinal complications interfere with the necessary long-term observation to determine the biocompatibility and the functional integrity of the device and the eye in vivo [1, 2]. The visualization of anatomical structures of the eye is usually done with 2D images of microscopic sections at the area of interest. Less often entire sections of the eye are presented. Often such images show artifacts such as retinal detachments or outer wall deformations. Such artifacts are likely because of processing related tissue reactions, e.g. fixation and dehydration. For a better understanding of the anatomy of the eye 3D visualizations may be helpful. They can be especially helpful in planning of difficult surgical procedures such as the implantation of complex structures and devices such as electronic retinal implant systems. They can also be helpful to facilitate the development and design of instruments and implants. 3D reconstructions can also enable us to practice difficult surgical procedures before animal experiments are performed. Eye surgery in humans can be simulated in virtual reality surroundings such as the EYESI® simulator. Simulator based surgical training seemed to be helpful for beginners in eye surgery [3–9]. However, this system is very expensive and restricted to a human eye model.

In this paper we demonstrate the difficulties in obtaining a 3D reconstruction of the eye and its internal structures by straightforward image stacks constructed from registered serial sections of eyes of mice, rats, and rabbits. A workflow is then described which can be used to calculate and visualize pseudorealistic 3D models of eyes of any species based on a single sagittal microscopic section of the eye and how one can integrate such models in different 3D viewing environments (Fig 1). This workflow is based on the simplified assumption that a sagittal section of the eye rotated around the axis through the corneal apex and the posterior pole of the retina delivers voxel data of a pseudorealistic 3D model suitable for surgery planning, training, and device development.

## Materials and methods

### Animals

We used five 15–28 weeks old C57BL/6J wild type (wt) mice (Janvier, Le Genest-Saint-Isle, France), four 7–8 weeks old Brown Norway rats (Institute of Laboratory Animal Science, Medical Faculty, RWTH Aachen University, Aachen, Germany), and three 7–8 weeks old

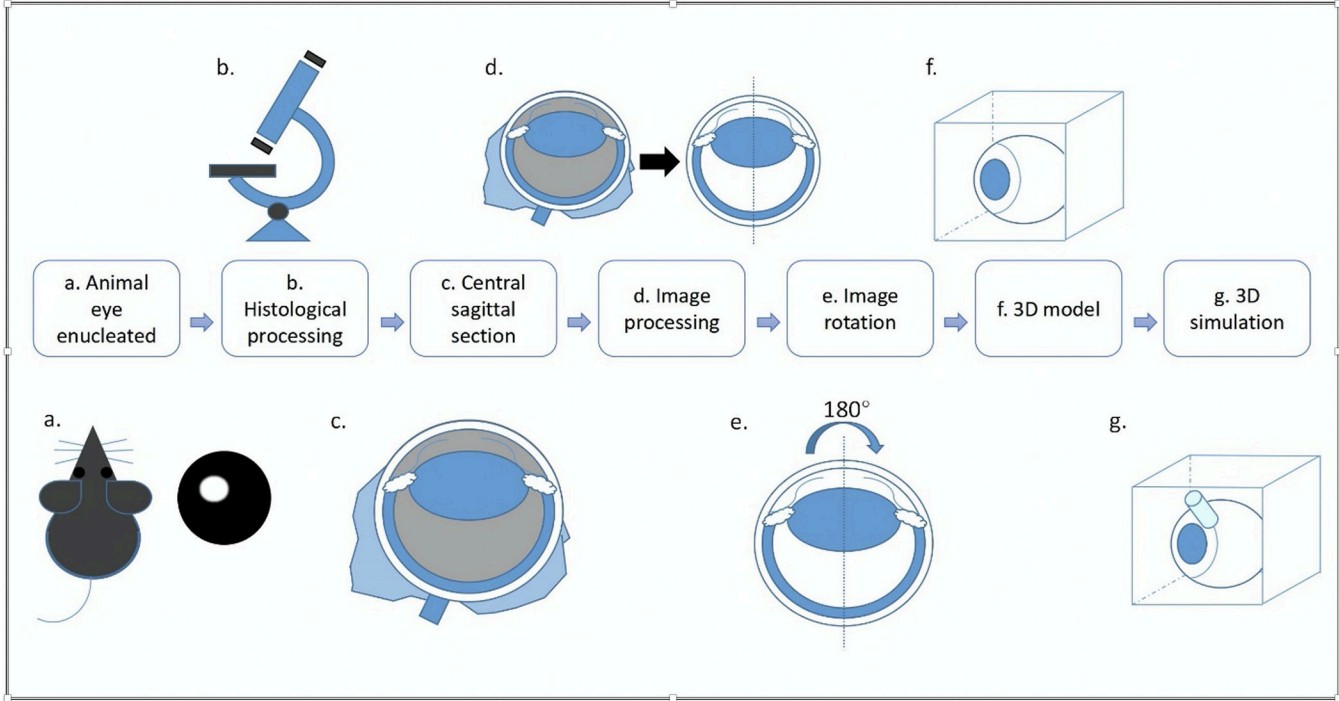

**Fig 1. Workflow of the 3D reconstruction from the rotation of single histological section.** (A) An eye was enucleated from one animal. (B) The eye was processed with standard histology. (C) The eye was sectioned after being embedded in paraffin. A central sagittal section was selected from the serial section stack and scanned. (D) Irrelevant tissue was eliminated from the image. The anterior chamber and the vitreous should be emptied and cleared by eraser functions of the image processing software. (E) The image was rotated 180˚ around the y-axis. (F) A 3D model of the eye was established. (G) The eye model was imported into a virtual reality (VR) platform and moving instruments were inserted to simulate surgical interventions.

Chinchilla-Bastard rabbits (Charles River, Châtillon, France). The animals were kept under controlled light conditions (twelve-hour light / twelve-hour dark cycle). Water and food were available ad libitum. For sacrifice, mice and rats were deeply anesthetized with isoflurane (Forene®, AbbVie, Wiesbaden, Germany) and killed by decapitation. Rabbits were killed with an overdose of 2 mL/kg bodyweight pentobarbital-sodium (Narcoren, 160 mg/mL; Boeh-ringer Ingelheim, Ingelheim am Rhein, Germany). All experiments were performed after approval was obtained by the regulatory authorities (40130A4 and 81–02.04.2021.A295) and in accordance with the ARVO statement for the Use of Animals in Ophthalmic and Vision Research and the German Law for the Protection of Animals.

## Histology

After euthanization of the animals the eyeballs were enucleated. Mouse and rat eyes were fully fixed in Methacarn [10] for 24 hours at 4.0˚C. In rabbit eyes Methacarn was firstly injected into the vitreous cavity after paracentesis and then fixed in the same solution for 24 hours at 4.0˚C. After dehydration, all eyes were paraffin embedded and 3.0 μm thick sections were serially cut using a microtome (Slide 4003E, PTM Medical, Cologne, Germany). These sections were stained with hematoxylin and eosin (HE) following deparaffination. The sections of mouse and rat eyes were examined and digitized using a microscope (Leica DM6000 B with 3CCD Digital Camera KY-F75U). Sections from rabbit eyes were scanned with a Slicescanner (PTM Medical, Cologne, Germany). In all cases, whole eye sections were processed.

**Table 1. Software required to perform the 3D reconstructions and for object visualization.**

| Software Title | Manufacturer | Websource | Comments |
|---|---|---|---|
| Photoshop | Adobe, San Jose, CA, USA | www.adobe.com | Processing of section photographs |
| Affinity Designer | Serif Europe, Nottingham, UK | www.affinity.serif.com | Processing of section photographs |
| MatLab | MathWorks, Massachusettes, USA | www.mathworks.com | Modell calculation and algorithm execution |
| Image J / Fiji | National Institutes of Health, USA | www.imagej.nih.gov/ij | Handling and visualizing of 3D image stacks, imaging of 3D objects with Volume Viewer, an Image J plugin |
| MeshLab | Visual Computing Lab, ISTI, Pisa, Italy | www.meshlab.org | Processing of voxel data and point clouds, surface reconstruction |
| Blender | Blender Foundation | www.blender.org | Virtual Reality environment, processing and visualization of 3D objects |

## Data processing

The required software is listed in Table 1.

## Original image stack visualization

Each slice obtained by serial sectioning of the eye was processed using object identification and centration, filtering to erase any unnecessary image information, and image alignment using translation, rotation, and scaling functions. The resulting image stacks were then imported into *Image J* (*Fiji*) and visualized with the plugins *3D Viewer* and *Volume Viewer* (see Table 1 for software details).

## Virtual image stack generation

Any image processing software can be used to process section photographs of the eye anatomy. For this project we worked with *Adobe Photoshop* and *Affinity Designer*. *MatLab* was used for the calculation of the voxel data sets. *Image J* was used to visualize 3D image stack data and object volume data. *MeshLab* was used to reconstruct a surface model of the eye [11]. *Blender* was used as a standard virtual reality (VR) environment for further visualization and complex scene generation and editing. The 3D reconstruction is based on a simple rotation of a symmetric sagittal section through the center of the eye. To make sure that the rotation around the y-axis of the eye is working images must be processed. The image must be arranged with the corneal apex positioned at the top edge of the image as shown in Fig 2A. The anterior chamber and the vitreous should have been emptied and cleared using image processing software to prevent artifacts in the 3D reconstruction (Fig 2B). Frequently, the sections were not symmetrical. In such cases the image was split along the y–axis with one half being duplicated, mirrored, and positioned alongside the first half so that a fully symmetrical image of the whole globe resulted (Fig 2C). In these cases, the representation of the optic disk was affected which was insignificant for the purpose of this study. The anterior chamber and the vitreous should have been emptied and cleared using image processing software to prevent artifacts in the 3D reconstruction. For practical reasons (processing time) the images should not exceed 600 x 600 pixels. The *MatLab* script analyzed the size of the image and in case it was larger than 300 pixels on the x-axis a resize command was executed resulting in down-sampling of the image. 300 x 300-pixel images were here used as a standard.

The 3D image data set was generated by a rotation of this section around the y-axis as shown in Fig 3 and a simple trigonometric projection. Each pixel of the original 2D image was given by its coordinates x and y and by its RGB color information. Rotating this image around the y–axis with an angle of led to a voxel with the same color information and with the same y-

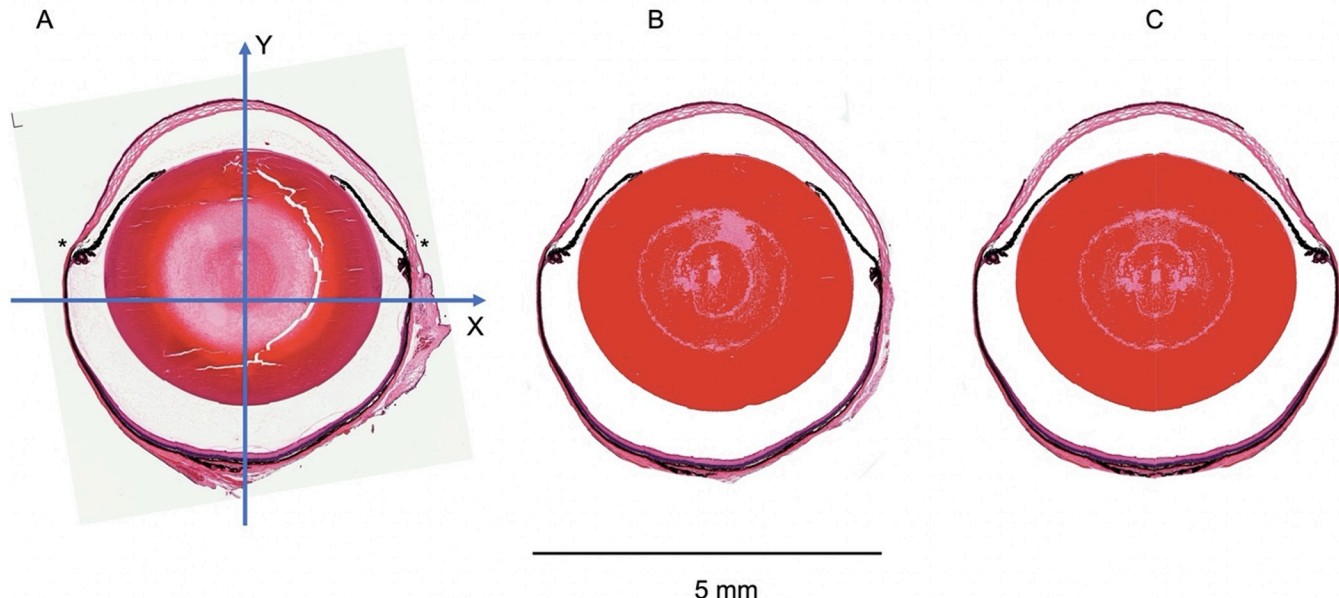

**Fig 2. Image processing of sagittal HE stained whole globe microscopic sections.** (A) The image was rotated so that the x -axis was defined as a horizontal line through the image center parallel to the corneoscleral junction (*). The y-axis was defined as a perpendicular line to the x-axis from the corneal apex to the posterior pole. (B) The vitreous cavity and the anterior chamber were cleared as well as adjacent tissue to the globe. Cracks in the lens structure were filled up. (C) For the following rotation procedure the image in (B) was cut vertically in two halves and the left image was duplicated, mirrowed, and added to the original left half so that a fully symmetrical image was obtained.

coordinate but with a new x-coordinate which was calculated based on the cosine of, and with a z-coordinate based on the sine of The voxel calculation was repeated for each xy-pixel of the original 2D image and for ranging from 0˚ to 180˚ in 0.2˚ steps. This calculation loop represented the core of the *MatLab* script described below. The voxel set being calculated from a

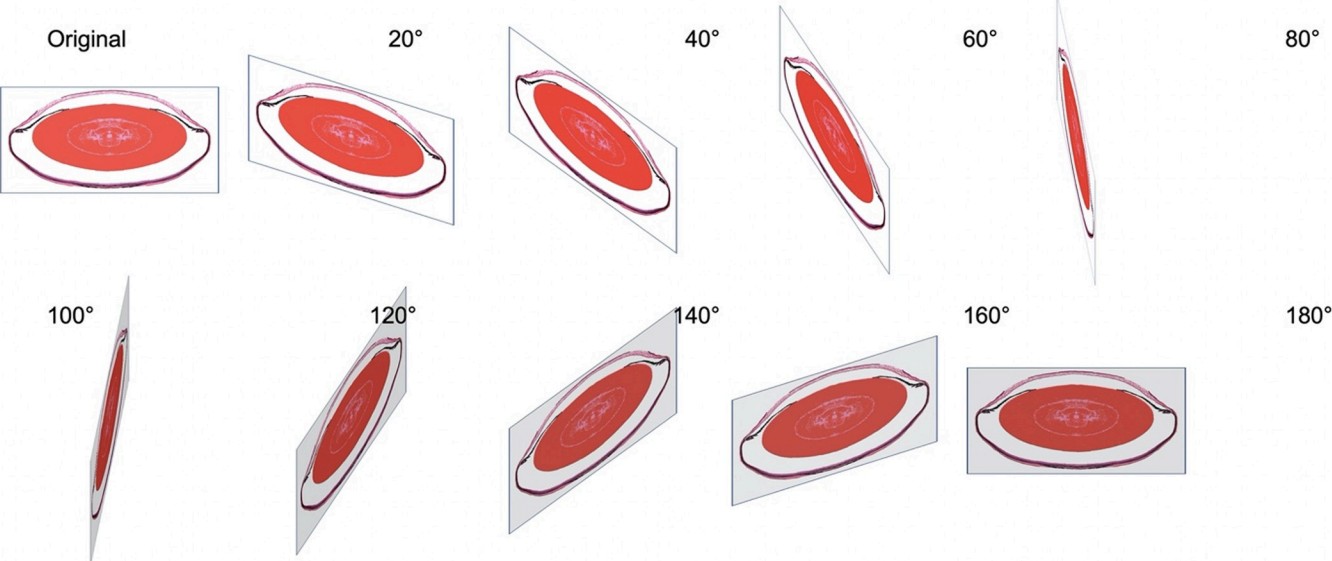

**Fig 3. Image rotation to create a 3D eye model.** Stepwise 20˚ rotations of an image showing a sagittal section of a mouse eye around the y-axis to obtain voxel data in the 3D domain from a simple 2D image based on the simplification that the eye is a rotational symmetric spherical object. The voxel coordinates were calculated using simple trigonometric projection rules. The 20˚ steps were used here to explain the principle. For real calculations 0.2˚ steps were used.

single 300 x 300 pixel image with a rotational resolution of 0.2˚ was based on a total number of 300 x 300 x 900 = 81,090,000 sine resp. cosine calculations. When testing these scripts, we usually worked with 5˚ stepwise rotations in order to speed up the visualization of the results. To achieve higher resolutions as demonstrated here, we used 0.2˚ steps for

The rotation algorithm for the generation of a stack of images presenting microscopic slices of the eye around the y-axis is shown in Table 2.

The resulted image stacks contained virtual slices which could then be imported to *Image J* and visualized with its *Volume Viewer* plugin.

## Generation of a point cloud

Point clouds are lists of voxel data of a 3D object with the information on the x-, y-, and z-coordinates and the color information of each point. Point clouds could be calculated from the rotational algorithm as described above or they could be extracted from the virtual image stack generated with the method described above. We used a short *MatLab* script to import the images from the stack and to extract the voxel list.

## Processing of point clouds

This resulting point cloud file was imported as a mesh into *MeshLab* [11]. The rotation of a 300 x 300 pixels image generated a point cloud of approximately 6 million vertices. With *MeshLab* filters then the normals were calculated to each of the vertices with the parameter "Neighbour Num" set to 100 and "Smooth iteration" set to 0. Then *MeshLab* generated the surfaces of the mesh using the *Surface Reconstruction Tool* with the *Screened Poisson Algorithm* according to Kazhdan and Hoppe [12]. Surfaces were then smoothed using the *MeshLab* filter *Laplacian Smooth* which was described by Olga Sorkine [13]. The resulting object could be viewed in *MeshLab* from different angles and distances. However, to use the object in complex scenes or to simulate object behavior an integration in a virtual reality platform seemed to be necessary. Therefore, the *MeshLab* object was stored as a Collada File indicated by the suffix '.dae' and then imported into *Blender*, an open-source virtual-reality (VR) platform. In *Blender* we were able to move the eye object, to rotate it, to scale it, or to combine it with other objects such as virtual surgical instruments.

## Biometry from HE stained histological sections

Corneal diameter, ocular axial length and retinal thickness were measured on scans of histological images. Retinal thickness was determined in sections at six positions close to the papilla, maximum distance of 0.6mm from the optic nerve head for rat and mouse eyes, and 3.0mm for rabbit eyes, respectively. These measurements were averaged to produce the recorded value. Values are given as means ± SD. For retinal thickness and ocular axial length, in-vivo values were cited from the literature [14–22].

## Results

### Histology

We performed serial sections of eyes of C57BL/J6 wt mice, brown Norway rats, and Chinchilla Bastard rabbits and selected the best central sagittal section for the reconstruction. Examples are given in Fig 4. All structures were well preserved, however typical artifacts occured, such as breaks in the lens or shallow retinal detachments. In rabbit eyes, we observed deformations of the globe (Fig 4C). For a biometrically correct reconstruction of the true anatomy, it is important to know the effect of processing-induced shrinkage. The results are shown in Table 3.

**Table 2. *MatLab* script executing the rotation algorithm for the generation of a stack of images presenting microscopic slices of the eye around the y-axis.**

| Command Lines | Comments |
|---|---|
| *% read image data (bmp, jpg, png) to IMG*<br>IMG = imread('maus.jpg'); | Read the image data. In this case a sagittal section of a mouse eye with the name 'maus.jpg' is loaded. |
| *%determine the size of image I*<br>*% with x length in S(1) and y length in S(2)*<br>S = size(IMG)<br>*% aspect ratio*<br>ar = size(IMG,1)/size(IMG,2);<br>*% resize IMG to x = 300 y = 300/ar*<br>iy = int16(300/ar);<br>I = imresize(IMG,[300 iy])<br>imshow(I)<br>S = size(I) | Resize image to 300 pixels on the x-axis preserving the aspect ratio with the y-axis. |
| *%preallocation of sphere_R/G/B (S(1),S(2),S(2))*<br>sphere_R = zeros(S(1),S(2),S(3));<br>sphere_G = zeros(S(1),S(2),S(3));<br>sphere_B = zeros(S(1),S(2),S(3)); | Preallocation of variable space of matrices and vectors reducing calculation time. |
| *% rotate 2D Image matrix I around the symmetry axis leading to a 3D matrix*<br>*% rotate*<br>rad = int16(S(2)/2)+1<br> for i = 1:S(1)<br> for j = 1:S(2)<br> for al = 0:0.2:180*%usually step 0.2*<br> sp_y = i;<br> sp_x = int16(cosd(al)*(j-rad))+rad+1;<br> sp_z = int16(sind(al)*(j-rad))+rad+1;<br> sphere_R(sp_x,sp_y,sp_z) = I(i,j,1);<br>sphere_G(sp_x,sp_y,sp_z) = I(i,j,2);<br>sphere_B(sp_x,sp_y,sp_z) = I(i,j,3);<br>*%sphere_R contains RED values*<br>*%Sphere_G contains GREEN values*<br>*%sphere_B contains BLUE values*<br> end<br> end<br>int16(i/S(1)*100)<br>end<br>*% result of 3D picture is stored in sphere_R(x,y,z), sphere_G, sphere_B for red, green and blue channels* | Rotation algorithm: The resulting voxel data set is saved in a set of three variables for each RGB channel.<br>The 0.2 step can be changed to larger values to speed up the visualization but with a loss of resolution. |
| *% reconstruct slices in z axis from RGB channel 3 D set*<br>for stack = 1:int16(size(sphere_B,3)/2)<br>image_R = sphere_R(:,:,stack);<br>image_G = sphere_G(:,:,stack);<br>image_B = sphere_B(:,:,stack);<br> for x = 1:size(sphere_B,1) for y = 1:size(sphere_B,2)<br> IM(x,y,1) = image_R(x,y);<br> IM(x,y,2) = image_G(x,y);<br> IM(x,y,3) = image_B(x,y);<br> end<br>end | Stack generation. |
| IM = uint8(IM);<br>*%RGB Bild in IM*<br>*% jetzt IM als Folge darstellen und abspeichern in A (x,y,color,stack position) as sequential images*<br>imshow(IM,'InitialMagnification',500)<br>A(:,:,:,stack) = IM;<br>end | Showing each stack element as a sequential image. |

*(Continued)*

**Table 2.** (Continued)

| Command Lines | Comments |
|---|---|
| *% eliminate black pixels in A*<br>si = size(A)<br> for i = 1:si(1)<br> for j = 1:si(2)<br> for k = 1:si(4)<br> if A(i,j,1,k)+ A(i,j,2,k)+ A(i,j,3,k) = = 0<br> A(i,j,1,k) = 255;<br> A(i,j,2,k) = 255;<br> A(i,j,3,k) = 255;<br> end<br> end<br> end<br>end | This routine eliminates pixels which are fully black. |
| *% Plot and save A as a sequence of RGB images*<br>for i = 1:int16(size(sphere_B,3)/2)<br> imshow(A(:,:,:,i))<br> filename = 'stack_pic'+string(i)+'.jpg';<br> imwrite (A(:,:,:,i),filename);<br> filename = 'stack_pic'+string((size(sphere_B,3))-i+1)+'.jpg';<br> imwrite (A(:,:,:,i),filename);<br>end | Now the corrected sequential images of the 3D stack are shown as a video and each image representing a virtual slice of the eye around the y-axis is saved to disk. |

After standard histological processing, a decrease in ocular dimensions was observed to varying degrees as detailed in Table 3 and Fig 5.

## 3D reconstruction of original serial sections

Serial sections of eyes were stored as image stacks after image alignment of each image to reduce the amount of positional and rotational errors. Fig 6 demonstrates a 3D view of a mouse eye generated from the original aligned image stack. It became obvious that artifacts from single images considerably reduced the quality of the obtained 3D view of the eye and that even after alignment the 3D model did not show a smooth organ model.

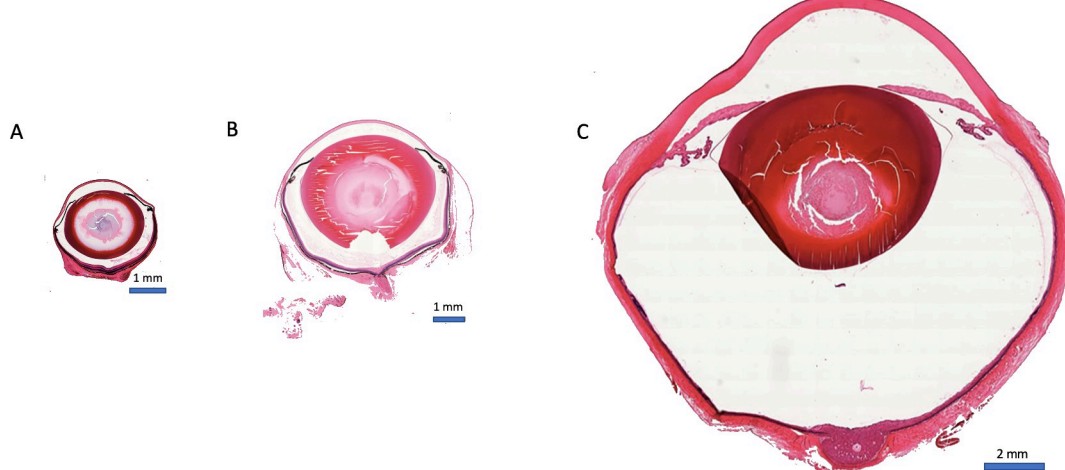

**Fig 4. Whole eye histology.** Examples of sagittal sections of (A) C57BL/6J wt mouse eye, (B) rat eye, and (C) rabbit eye. Whole globes were harvested and processed for hematoxylin and eosin staining. Magnification, 20X. Horizontal size indicator: 1 mm in (A) and (B), 2 mm in (C). Slices were scaled for size comparison.

**Table 3. Ocular dimensions and effects of histological processing.** Mean values ± standard deviation. First line represents data in-vivo with the ocular length and central retinal thickness data taken from the literature; second line shows data from histological sections; third line gives the mean shrinkage factor as the ratio between the post-processing mean and the in-vivo mean, respectively. Superscript numbers refer to the literature source of the data, unreferenced data represent own measurements.

| parameter | mouse (n = 5) | rat (n = 4) | rabbit (n = 3) |
|---|---|---|---|
| Corneal diameter | 2,70±0.24mm<br>2.42±0.11mm<br>0.896 | 5.50±0.0mm<br>4.80±0.28mm<br>0.873 | 13.63±0.41mm<br>11.79±0.68mm<br>0.865 |
| Ocular axial length | 3.00±0.04mm [14,15,18]<br>2.77±0.18mm<br>0.923 | 6.29±0.26mm [19,20]<br>5.13±0.30mm<br>0.816 | 17.10±0.41mm [21,22]<br>14.68±0.36mm<br>0.858 |
| Central retinal thickness | 186.90±15.10µm [14,15]<br>172.95±28.0µm<br>0.925 | 199.2±7µm [16]<br>144.82±24.40µm<br>0.727 | 194.30±7.70µm [17]<br>160.39±33.34µm<br>0.826 |

## 3D reconstruction of a pseudorealistic eye model

The rotational model based on a single symmetric sagittal section through the eye as explained above yielded a stack of virtual slices through the eye which can be used for visualization of the eye as demonstrated in Fig 7 showing a mouse eye.

For more complex visualizations VR platforms and scene editors are more useful. The application of such software platforms required the generation of surfaces from the voxel data.

A pointcloud as shown in Fig 8 was calculated by a *MatLab* script as described above and uploaded as a PLY file to *MeshLab* with the command IMPORT MESH from the FILE menu.

In *MeshLab* the first step was to reduce the number of voxels or vertices if they exceed more than 3 million in order to reduce processing time. This was done by the Command POINT CLOUD SIMPLIFICATION. Then, after the calculation of Normals and the Surface Reconstruction by a Screened Poisson technique, *MeshLab* presented the 3D object consisting of a large number of triangular faces. Fig 9 shows an example of the triangular surface elements in this case of the posterior part of the lens of a mouse eye. Examples of the 3D view of a rat eye provided by *MeshLab* from different angles and distances are shown in Fig 10.

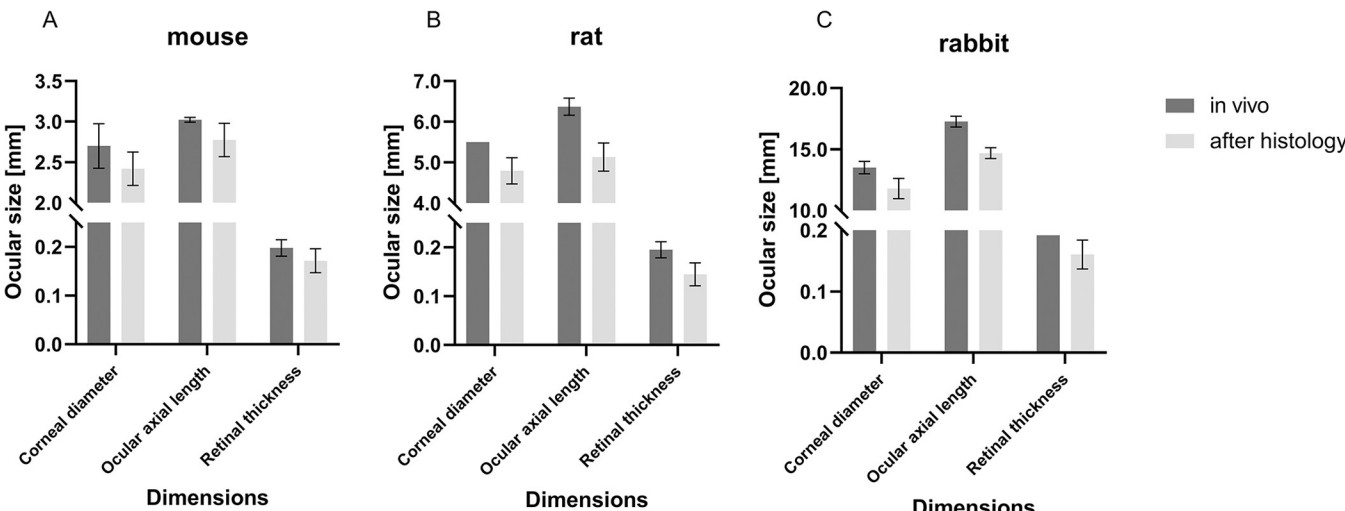

**Fig 5. Biometric effects of histological processing.** Ocular dimensions and processing induced size reduction of mouse eyes (A), rat eyes (B) and rabbit eyes (C). Data presented as means ± SD.

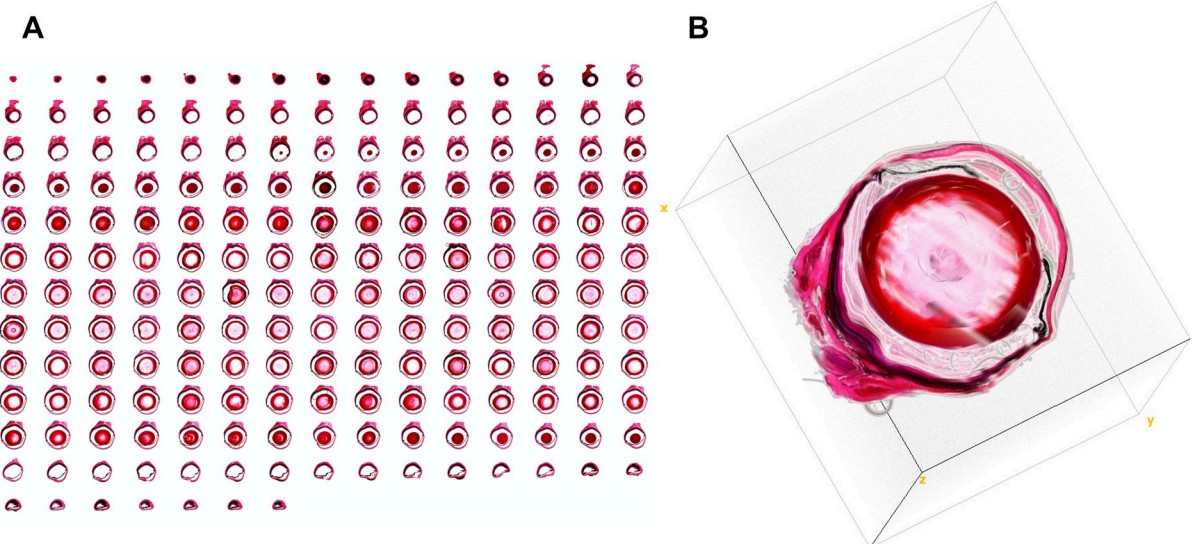

**Fig 6. Image stack based mouse eye reconstruction.** Example of a 3D reconstruction of a mouse eye based on an original image stack preprocessed with image alignment. (A) Montage of single slices. (B) 3D reconstruction generated by the *Volume Viewer* Utility of *Image J*.

## Import of eye objects into a Virtual Reality (VR) platform

To use the VR 3D reconstruction in a scene builder and to include moving instruments, the surface model had to be imported in a VR surrounding environment such as *Blender*. This was be done by exporting the mesh from *MeshLab* as a Collada file (.dae file). Fig 11 shows examples of eyes of a mouse, a rat, and a rabbit with a realistic size ratio together with virtual

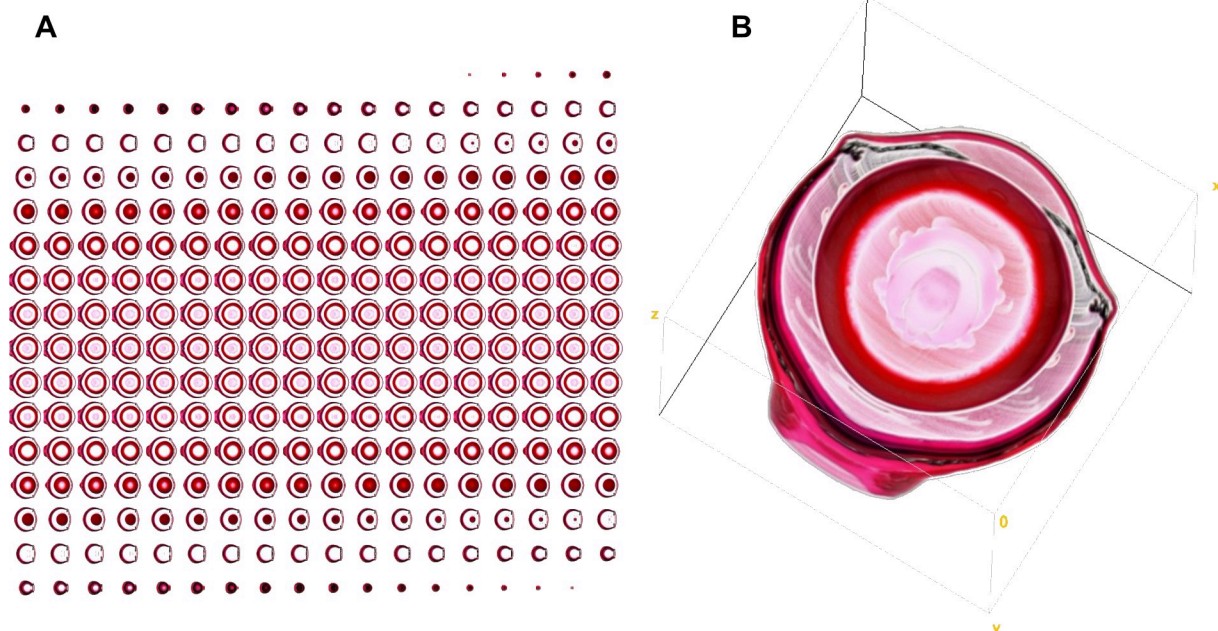

**Fig 7. Virtual image stack based mouse eye reconstruction.** Example of a 3D reconstruction of a mouse eye based on an image stack created by virtually sectioning the 3D voxel cloud as calculated based on a y-axis rotation of a sagittal section. (A) Montage of all slices generated by the *MatLab* script. (B) Visualization of the image stack performed with '*Volume Viewer*', an *Image J* plugin.

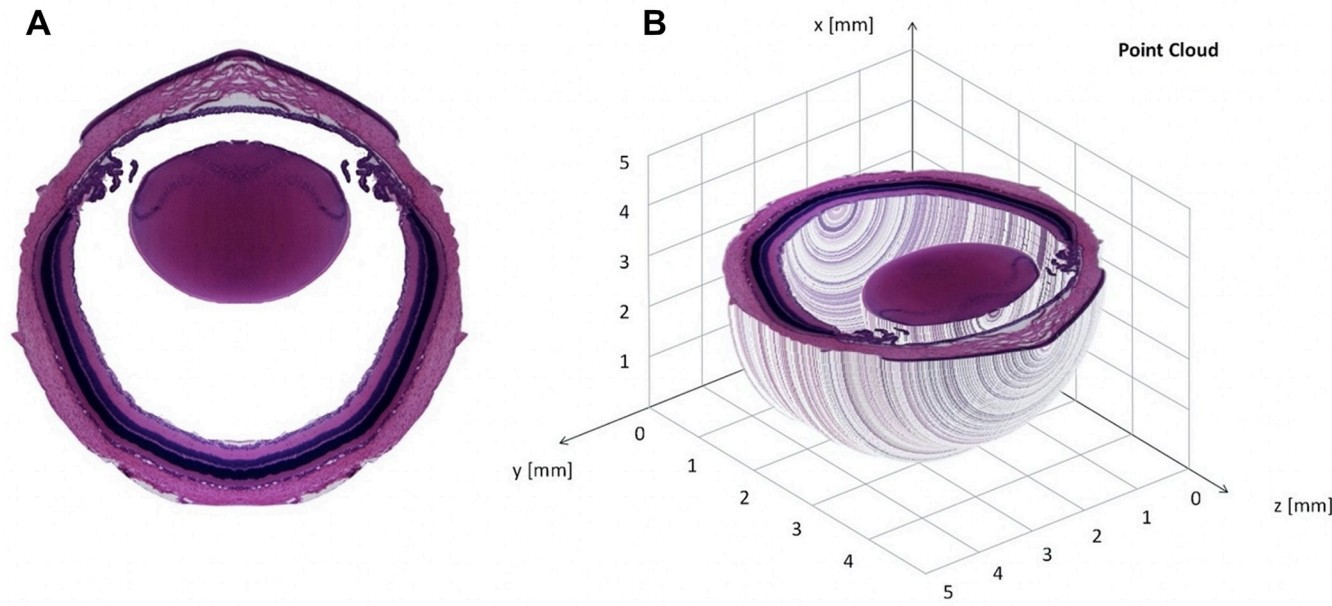

**Fig 8. Pointcloud of a rat eye.** (A) Preprocessed symmetric image of a rat eye from one sagittal section with the vitreous and the anterior chamber cleaned by image processing software. (B) Resulted point cloud as calculated with the *MatLab* script given in Table 4.

**Table 4.** *MatLab* script to extract the point cloud data from a stack of images.

| COMMAND LINES | COMMENTS |
|---|---|
| % STACK TO POINT CLOUD<br>% preallocation sphere data | |
| IMG = imread('stack_pic50.jpg')<br>S = size(IMG);<br>sphere = zeros(S(1),S(2),S(1)-1,3); | Read one image from the stack to determine its size and preallocate the sphere date to accelerate the calculations |
| for z = 1: S(1)-1<br>datei = "stack_pic"+string(z)+".jpg";<br> I = imread(datei);<br><br> for x = 1: S(1)<br> for y = 1:S(2)<br> sphere(x,y,z,1) = I(x,y,1);<br> sphere(x,y,z,2) = I(x,y,2);<br> sphere(x,y,z,3) = I(x,y,3);<br> end<br> end<br>end | Read each image from the stack and import the x,y,z coordinates as RGB data |
| % calculate point cloud<br>% preallocation of maximum possible size of coordinate matrix<br>xyz = zeros(S(1)*S(2)*(S(1)-1),3);<br>%preallocation of maximum possible size of color matrix<br>col = zeros(S(1)*S(2)*(S(1)-1),3,'uint8'); | Preallocation of coordinate and color matrix |
| BP1 = 5;<br>BP2 = 230; | Bandpass filter for color information |

(*Continued*)

**Table 4.** (Continued)

| COMMAND LINES | COMMENTS |
|---|---|
| k = 0;<br><br>for z = 1:size(sphere,3)/2%fast step 5<br> z<br> for y = 1:size(sphere,2)<br> for x = 1:size(sphere,1)<br><br> if sphere(x,y,z,1)>BP1<br> if sphere(x,y,z,1)< BP2<br> k = k+1;<br> xyz(k,1) = x;<br> xyz(k,2) = y;<br> xyz(k,3) = z;<br> col(k,1) = sphere(x,y,z,1);<br> col(k,2) = sphere(x,y,z,2);<br> col(k,3) = sphere(x,y,z,3);<br> end;<br> end;<br><br> end;<br> end;<br> z<br>end; | Calculate list of coordinates *xyz* and colors *col* (k counts the number of valid and non zero data)<br>End z-loop at 'size(sphere,3)/2' gives a 180˚ rotation where as 'size(sphere,3)' gives a 360˚ rotation |
| ptCloud = pointCloud(xyz,'Color',col);<br>pcshow(ptCloud);<br>pcwrite(ptCloud,'test_pc','PLYFormat','binary');<br>pcwrite(ptCloud,['test_pc.pcd'],'Encoding','ascii'); | Write point cloud to files "test_pc.ply" and "test_pc.pcd" |

instruments of correct size ratios inside the eye. By moving and rotating the virtual instruments the range of possible movements without collision to the eye tissue and specific components could be explored.

## Discussion

3D visualization of human organs or body parts are already used in disciplines such as neurosurgery [23], orthopedic surgery [24], dental and maxillofacial surgery [25] for navigation, implant positioning, and also for individualized fabrication of custom designed implants. These systems are usually based on 3D data sets provided by Computer Tomography scanning techniques, Magnetic Resonance Imaging data sets, and other imaging methods such as ultrasound. 3D data of the human eye can also be obtained by these techniques as shown by Daftari et al in the case of uveal melanomas imaged by 3D Magnetic Resonance Imaging for planning of proton beam therapy [26]. However the spatial resolution is low when the demands of ophthalmic microsurgery is considered. 3D Navigation using such data sets have been used in orbital decompression surgery, or in the surgical placement of tantalum clips for proton beam irradiation [27, 28]. This approach was only used for extraocular surgery, and a spherical geometric model of the eye is used. To our knowledge a realistic 3D model of whole eyes including tissue differentiation and microscopic resolution based on light microscopy data has not been established. Such a model would be helpful for designing of new devices, implants, and surgical procedures. An approach with a different intention is the use of confocal images fitted into a geometrical model of an eye supplemented with photorealistic visuals [29]. This concept is very helpful for education. In contrast, a straightforward approach with serial sections of the globe and visualization of the image stack with 3D viewing software remains difficult even after alignment and preprocessing of each stack slice due to artifacts resulting from the

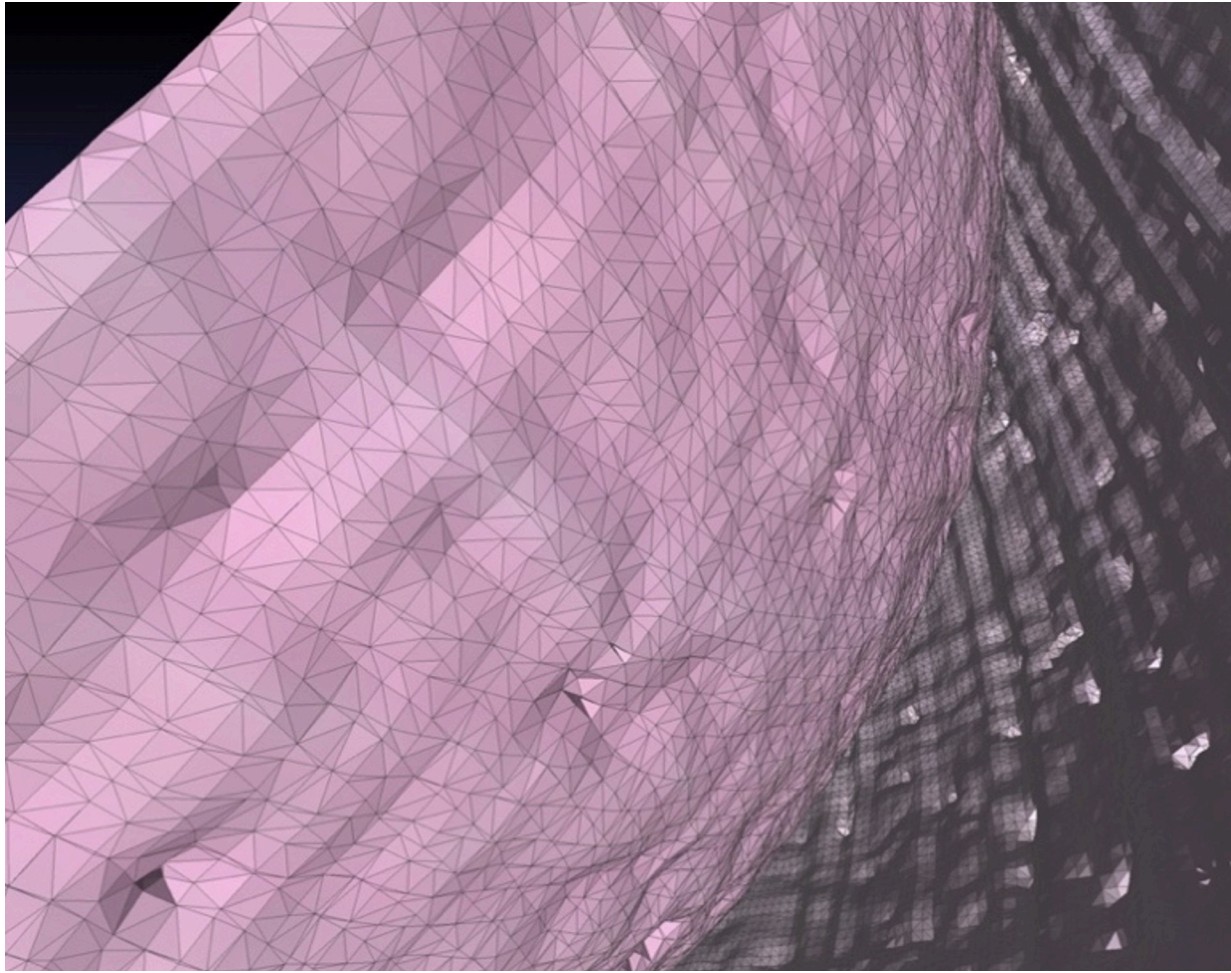

**Fig 9. Example for surface reconstruction.** Surface reconstruction of the lens of a mouse eye using *MeshLab* based on the PLY point cloud as calculated by the script in Table 4. Triangular faces as a result of the rated poisson reconstruction are indicated and given. In this example 2.27 Mio triangular faces or surface elements were calculated for the mouse eye.

fixation, embedding, and slicing process such as deformations, disruption of the tissue, tissue folds, and tissue losses. We therefore created a protocol to visualize a pseudorealistic 3D model of eyes of different laboratory animals with microscopic resolution and tissue differentiation for preclinical planning and simulation of eye surgeries and device development. Eyeballs undergoing processing of standard histology showed shrinkage in different degrees which lead to a smaller image than in the in-vivo reality. Our results of retinal thickness for mice using histological sections tied well with previous studies of Rösch et al [30].

Non-invasive, slide-free imaging procedures may be helpful in avoiding artifacts and tissue damage. The application of such techniques is an important step forward in digital pathology [31]. In human examinations, 3D ultrasound imaging can already be regarded as a state-of-the-art technique [32]. For the purpose of this study we did not use ultrasound because of the lower spatial resolution compared to standard light microscopy. The resolution of conventional 10 MHz scanners is in the range of 0.45 mm lateral resolution and 0.15 mm axial resolution. Using 50 MHz scanners for ultrasound biomicroscopy enhances the resolution up to 60.0 μm and 30.0 μm, respectively but the imaging depth is currently limited to the anterior segment [33]. In ophthalmic imaging optical coherence tomography (OCT) is now widely

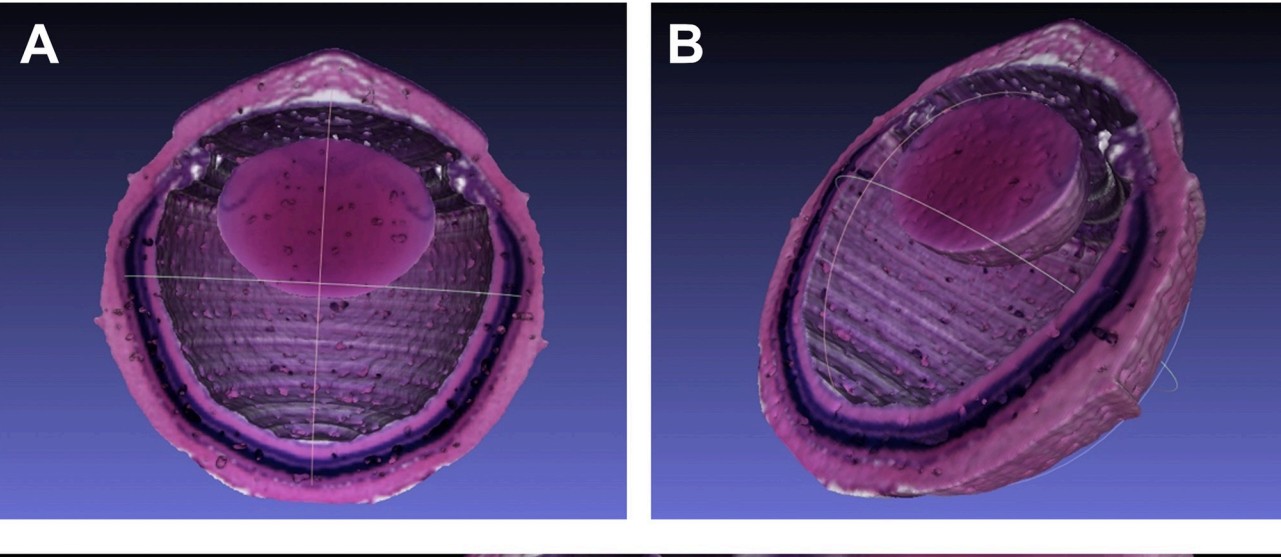

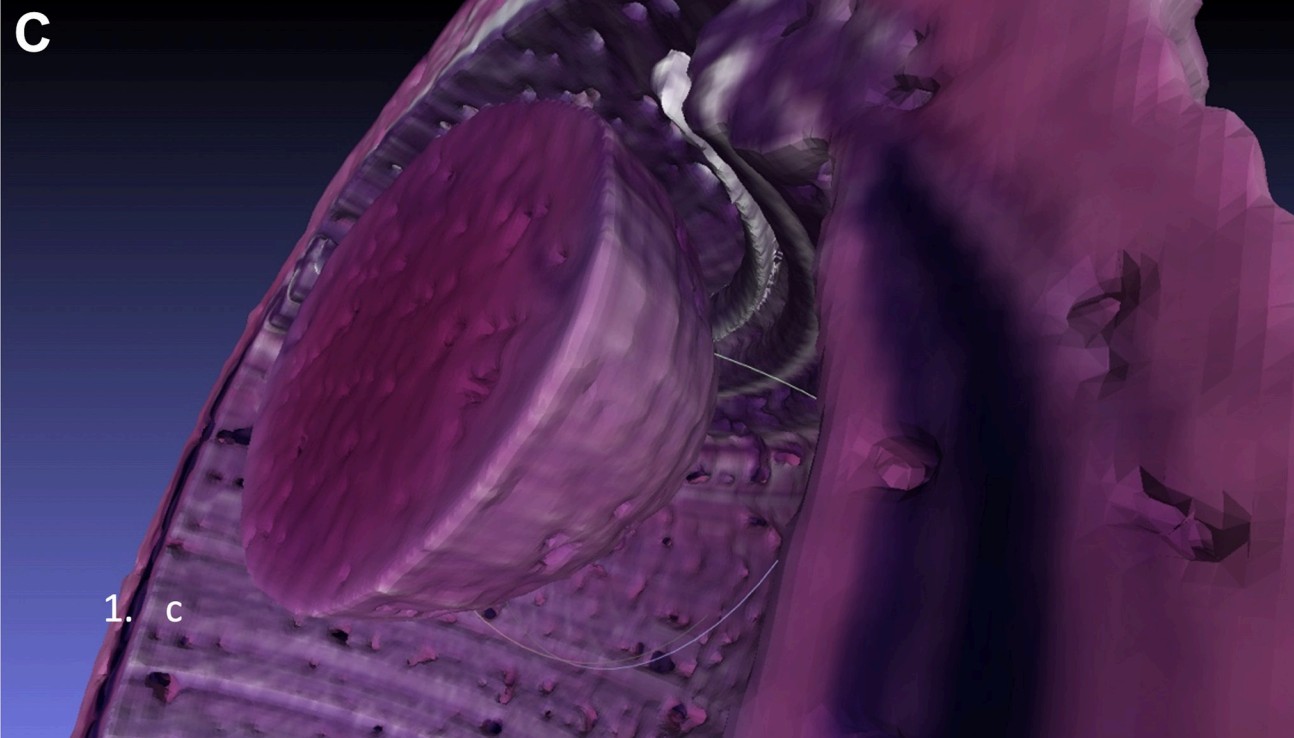

**Fig 10. 3D reconstruction of a rat eye in *Mesh Lab*.** (A, B) Half cut through the globe from different angles. (C) Detailed view from the space between the anterior retina and the lens.

used as a standard imaging technology for the retina as well as for the anterior segment with a resolution closer to that of light microscopy. Commercially available spectral domain OCT systems provide an axial resolution of 5.0 μm and a lateral resolution of 15.0 μm which can be achieved only for visible structures and the field of view is limited [34]. By combining anterior and posterior segment OCT scanning whole eye imaging can be done. The images taken by such systems show viable sagittal sections of an eye but the visualization of the area between the equator and the ciliary body, which is important for surgical procedures, remains difficult [35].

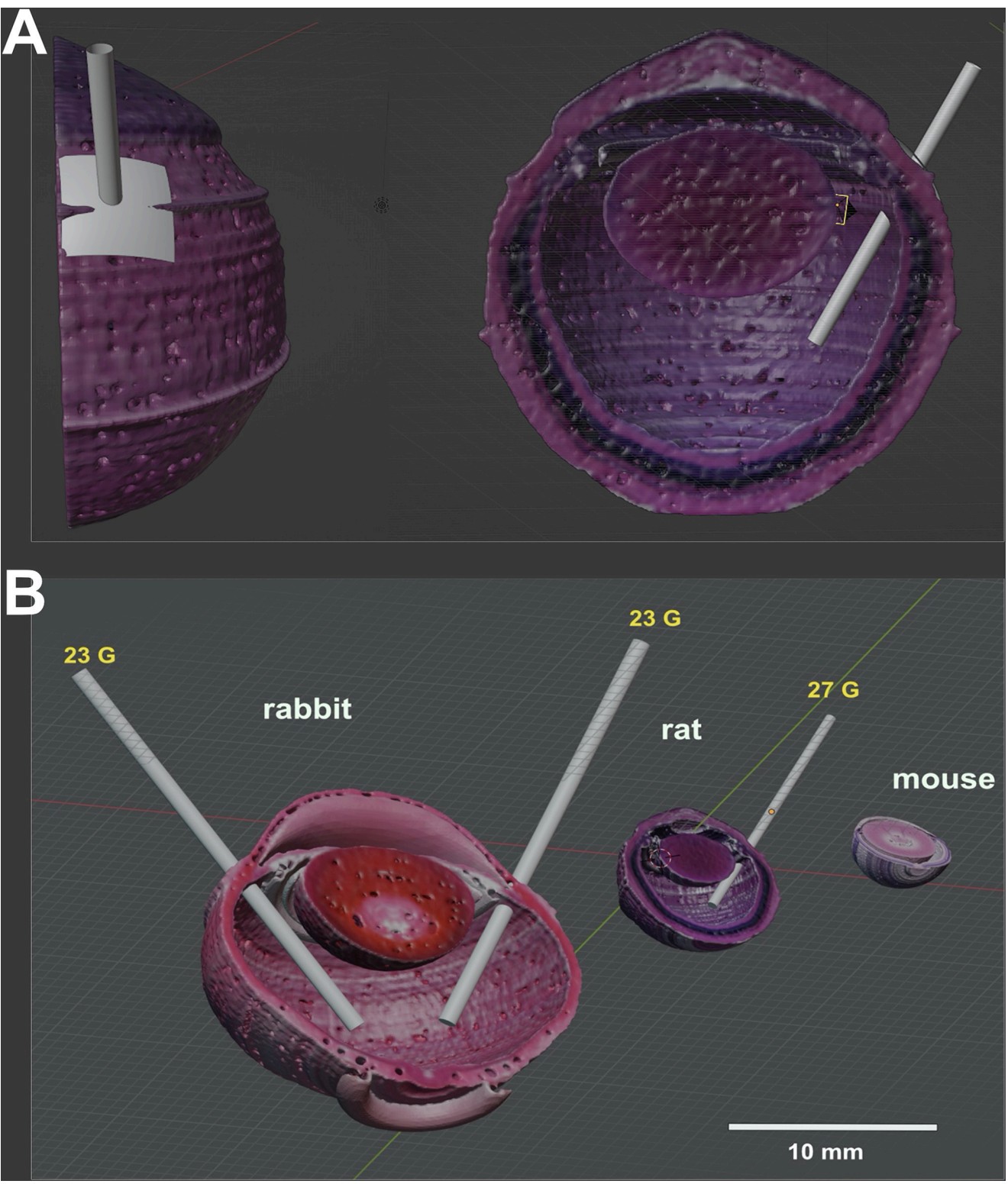

**Fig 11. 3D eye models in a VR scene building platform.** Eye reconstructions for surgical simulations created in the open source 3D scene editor software *Blender*: (A) Rat eye from different viewing angles with a surgical instrument inserted into the vitreous cavity. (B) Comparison of the rabbit, the rat, and the mouse eye with 23 gauge instruments in the rabbit eye, and a 27 gauge instrument in the rat eye.

Another non-sectioning technology to image organs in small animals is micro-computerized tomography (micro-CT) [36]. An advantage of this technique is that the 3D reconstruction is part of the regular workup for CT scanning and integrated in the imaging software. The spatial resolution of micro-CT seemed to be sufficient to visualize surgically important structures, but tissue differentiation may be problematic without staining. However, in future experiments using innovative staining agents we will use micro-CT for non-invasive imaging of small eyes. This technique could be extremely helpful in the preoperative planning before performing surgery in an individual animal. Recent studies of Runge et al. and Leszczynski et al. demonstrate the feasibility of this advanced technique [37, 38].

Another recent development in slide-free non-destructing light-microscopy is light-sheet fluorescence microscopy (LSFM) [39]. Usually, samples for LSFM are smaller than whole eyes but at least for the mouse and rat eye, LSFM is possible. Prahst et al. demonstrated slide-free 3D images of the retinal vasculature and especially subcellular structures and single neurons using LSFM [40]. We performed preliminary experiments in whole eyes without success because of the pigmentation of retinal cells interfering with the LSFM bleaching process. Obviously, for imaging of the eye, an optimized bleaching protocol needs to be established.

3D modeling of the eye in VR platforms is a wonderful tool helping to design new instruments, new devices, and new procedures. Also, training of novice eye surgeons is another possibility to use these data sets. Currently the EYESI® surgical simulator is the only available training system based on a human model eye providing a visual feed-back of the interaction between instruments inserted into the eye and the ocular structures along with the look and feel of a surgical microscope [9]. The workflow and software tools presented here may help to easily establish at least some opportunities to provide such a visual feedback and simulation of surgical procedures and instrument action in eyes of different species often used in experimental settings. In addition to surgical training with porcine cadaveric eyes or surgical eye models [41], this approach may also help to support the 3R principle of animal research, namely *Replacement*, *Reduction*, and *Refinement* proposed by Russel and Burch [42]. This concept is broadly considered as a systematic approach to animal experimentation that emphasizes laboratory animal welfare. Such a model-based strategy would minimize the number of animal experiments required for both, the investigation of compatibility and functionality and for the training of future surgeons. By reducing the number of complications of the insertion and placement of complex implants to a minimum, the number of animals used for feasibility trials can be reduced. The refined surgical procedure would prevent discomfort and reduces stress of the experimental animals.

In this first approach, we were not concerned with simulating biomechanical tissue properties, but only geometric-anatomical conditions and simulating surgical tactical spaces. However, for coming closer to a surgical VR model of different species, biomechanical characteristics of the ocular structures need to be determined and included into the model as well, e.g. data from Schwaner et al., who described biomechanical properties of the rat sclera using inverse finite element modeling [43].

The workflow as described here, covers a wide range of steps from enucleation and histological processing via image acquisition and processing to 3D reconstruction and the surgical simulation. Its benefit can also be found in non-ophthalmological areas of medical research where histological structures of human or laboratory animals play an important role. E.g. 3D reconstruction of anatomical structures contributed greatly to precise and safe execution of segmentectomy, for instance [44].

However, this study has its limitations. 1st, the basic very simplifying concept is that a full rotation of a sagittal section of the eye will result in an original model of the true anatomic situation of the eye which is certainly not true. However, based on the developmental process of

the eye being formed from a bleb of the neural tube it is at least a first approximation. 2nd, a relatively small number of experimental animals and species were included in the study, which affects the variability of quantitative data on biometry and shrinkage. 3rd, the obtained 3D model always belongs to an individual eye which was removed and processed for histology. Using such a model for a surgical simulation of a given eye in-vivo before the real surgical procedure poses the risk, that the anatomical situation may be slightly different due to individual variations. There can also be a difference between wild type animals and animals with retinal degenerations which are the focus of the development of retina implants. In future experiments we will therefore obtain data in animal models of such degenerations such as in the rd10 mouse. 4th, due to deformity of eyes after standard histological processing, 3D models generated by rotation of one single image may not be exactly the same as the in-vivo ones. In further projects also other non-destructive image processing approaches should be taken into consideration.

## Conclusions

In conclusion, we have described a workflow to calculate a pseudorealistic 3D model of eyes of small laboratory animals including tissue differentiation and light microscopic resolution. This model can be used in VR environments to simulate complex surgical procedures and it may support the development and design of ophthalmic implants such as retina implants. It will also help to prevent surgical complications in a very early stage of an experiment. Further work is needed to add mechanical properties to the 3D models for dynamic simulations and tissue–instrument–implant interactions.

## Acknowledgments

The histological processing was supported by the IZKF Aachen Core Facility Immunohistology. We thank Tim Gerrits and his team from the Virtual Reality and Immersive Visualization Team, RWTH Aachen, Germany for support with image processing, stack alignment and registration, and fruitful constructive discussions.

## Author Contributions

**Conceptualization:** Jiayun Wang, Sabine Baumgarten, Sandra Johnen, Peter Walter.

**Data curation:** Jiayun Wang, Sabine Baumgarten, Sandra Johnen, Peter Walter, Tibor Lohmann.

**Formal analysis:** Sandra Johnen.

**Funding acquisition:** Peter Walter.

**Investigation:** Jiayun Wang, Tibor Lohmann.

**Methodology:** Jiayun Wang, Sabine Baumgarten, Sandra Johnen, Peter Walter.

**Project administration:** Sandra Johnen.

**Resources:** Peter Walter.

**Software:** Frederic Balcewicz, Peter Walter.

**Supervision:** Sabine Baumgarten, Sandra Johnen, Peter Walter, Tibor Lohmann.

**Validation:** Jiayun Wang, Sandra Johnen, Tibor Lohmann.

**Visualization:** Jiayun Wang, Frederic Balcewicz, Peter Walter.

**Writing – original draft:** Jiayun Wang.

**Writing – review & editing:** Sabine Baumgarten, Frederic Balcewicz, Sandra Johnen, Peter Walter, Tibor Lohmann.

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
