## [Decision Letter · Decision Letter 0]

12 Jun 2023

PONE-D-23-15660A workflow to visualize vertebrate eyes in 3DPLOS ONE

Dear Dr. Walter,

Thank you for submitting your manuscript to PLOS ONE. After careful consideration, we feel that it has merit but does not fully meet PLOS ONE’s publication criteria as it currently stands. Therefore, we invite you to submit a revised version of the manuscript that addresses the points raised during the review process.

We look forward to receiving your revised manuscript.

Kind regards,

Shinji Kakihara, M.D.,Ph.D.

Academic Editor

PLOS ONE

Journal Requirements:

"all authors: GRK2610/1

DFG , https://www.dfg.de/

No, the funder did not play any role in the study."

"This work was supported by DFG grant GRK 2610/1.

The histological processing was supported by the IZKF Aachen Core Facility Immunohistology.

We thank Tim Gerrits and his team from the Virtual Reality and Immersive Visualization Team, RWTH Aachen, Germany for support with  image processing, stack alignment and registration, and fruitful constructive discussions. "

"all authors: GRK2610/1

DFG , https://www.dfg.de/

No, the funder did not play any role in the study."

"YW: no

SB: no

FB: no

SJ: no

PW: no

TL: no"

Additional Editor Comments:

Thank you for the opportunity to handle such a wonderful manuscript.

The manuscript entitled “A workflow to visualize vertebrate eyes in 3D” has been reviewed by two reviewers, and both this academic editor and two reviewers have a great interest in your manuscript.

However, the reviewers have also identified a few issues requiring clarification. Especially, as both reviewers commented, the conclusions are not based on the results and should be modified.

Therefore, we invite you to submit a revised version of the manuscript that addresses the points raised during the review process.

Reviewers' comments:

Reviewer's Responses to Questions

**Comments to the Author**

1. Is the manuscript technically sound, and do the data support the conclusions?

Reviewer #1: Partly

Reviewer #2: Partly

2. Has the statistical analysis been performed appropriately and rigorously? 

Reviewer #1: Yes

Reviewer #2: Yes

3. Have the authors made all data underlying the findings in their manuscript fully available?

Reviewer #1: Yes

Reviewer #2: Yes

4. Is the manuscript presented in an intelligible fashion and written in standard English?

Reviewer #1: Yes

Reviewer #2: Yes

5. Review Comments to the Author

Reviewer #1: The authors used image analysis software to construct a 3D imaging system for surgical procedures in small animal experiments, and it is exciting that they addressed the problem of artifacts in H&E-stained serial section images by creating masquerade images. However, several questions remain, which I would like to discuss.

1. Although this imaging system targets small animals, 3D images have already been used as preoperative data for patients with other organ diseases. Are there similar efforts in the field of ophthalmology? It should be discussed in this paper.

2. As the authors also mentioned in the Discussion, more detailed observation of internal structures is possible when ultrasound is used alone/combined. Is there a reason why it was not used in this experiment?

3. the authors state in their conclusion that it is indicated for future surgical experiments on small animals. However, all of the animals examined in this study were wild-type animals. There are limitations to using only masquerade images when actual experimental animals are used, as individual differences may occur strongly. It is recommended that the conclusion be changed and stated as a limitation.

Reviewer #2: Thank you very much for interesting paper about 3D modeling of experimental animals eyes.

I am impressed that your method is certainly useful in early-stage experimentation and could contribute to reduce the number of sacrificed animals.

Please refer the following comments.

Overall

1. Authors mention that original historical data have problems to create 3D model, and then use a single sagittal section to deal with problems. Please consider to show the comparison of 3D-models based on these two strategies in order to indicate their superiority or inferiority.

2. Maybe, your goal is to establish standard 3D-eye model for each experimental animal. Because your strategy is based on histological data, your model cannot be available for planning surgical procedures for individual subjects (modeling could not be prior to surgery). How about the variations in ocular dimensions of your subjects and deliverable 3D models? Are your models (Figure 7-11) available as standard models?

3. Authors analyzed the tissue shrinkage before and after histological procedures. Please show the actual data about ocular dimensions (corneal diameter, axial length, central retinal thickness) for clarity.

Abstract

4. Conclusion should be based on your results. Please consider to summarize your key messages for conclusion and add the data that support your conclusion.

Other minor comments

5. English editing is recommended to minimize redundancy in the manuscript.

6. PLOS authors have the option to publish the peer review history of their article (what does this mean?). If published, this will include your full peer review and any attached files.

Reviewer #1: No

Reviewer #2: No

---

## [Author Response · Author response to Decision Letter 0]

24 Jul 2023

Dear Reviewers and Editors, thanks for careful reading and and your comments and critics. We feel, that with your work we were able to improve the manuscript. All changes are highlighted in the file Revised manuscript with track changes and all changes and anwers are given in the file Response to reviewers. Thanks again best regards, Peter Walter on behalf of all authors

---

## [Decision Letter · Decision Letter 1]

8 Aug 2023

A workflow to visualize vertebrate eyes in 3D

PONE-D-23-15660R1

Dear Dr. Walter,

We’re pleased to inform you that your manuscript has been judged scientifically suitable for publication and will be formally accepted for publication once it meets all outstanding technical requirements.

Kind regards,

Shinji Kakihara, M.D.,Ph.D.

Academic Editor

PLOS ONE

Additional Editor Comments (optional):

Thank you so much for the revision.

I think the current version deserves publication in Plos One because the authors properly revised it according to reviewers' concerns.

Reviewers' comments:

Reviewer's Responses to Questions

**Comments to the Author**

1. If the authors have adequately addressed your comments raised in a previous round of review and you feel that this manuscript is now acceptable for publication, you may indicate that here to bypass the “Comments to the Author” section, enter your conflict of interest statement in the “Confidential to Editor” section, and submit your "Accept" recommendation.

Reviewer #1: All comments have been addressed

Reviewer #2: All comments have been addressed

2. Is the manuscript technically sound, and do the data support the conclusions?

Reviewer #1: Yes

Reviewer #2: Yes

3. Has the statistical analysis been performed appropriately and rigorously? 

Reviewer #1: Yes

Reviewer #2: Yes

4. Have the authors made all data underlying the findings in their manuscript fully available?

Reviewer #1: Yes

Reviewer #2: Yes

5. Is the manuscript presented in an intelligible fashion and written in standard English?

Reviewer #1: Yes

Reviewer #2: Yes

6. Review Comments to the Author

Reviewer #1: (No Response)

Reviewer #2: Thank you for showing me the revised manuscript. I think the authors have addressed my comments adequately.

7. PLOS authors have the option to publish the peer review history of their article (what does this mean?). If published, this will include your full peer review and any attached files.

Reviewer #1: No

Reviewer #2: No

---

## [Editor Report · Acceptance letter]

11 Aug 2023

PONE-D-23-15660R1 

A workflow to visualize vertebrate eyes in 3D 

Dear Dr. Walter:

I'm pleased to inform you that your manuscript has been deemed suitable for publication in PLOS ONE. Congratulations! Your manuscript is now with our production department. 

Kind regards, 

on behalf of

Dr. Shinji Kakihara 

Academic Editor

PLOS ONE